# Retrieving Proton Beam Information Using Stitching-Based Detector Technique and Intelligent Reconstruction Algorithms

**DOI:** 10.3390/s25164985

**Published:** 2025-08-12

**Authors:** Chi-Wen Hsieh, Hong-Liang Chang, Yi-Hsiang Huang, Ming-Che Lee, Yu-Jen Wang

**Affiliations:** 1Department of Electrical Engineering , National Chung Cheng University, Chiayi 621301, Taiwan; chiwenh@ccu.edu.tw (C.-W.H.); sam7987603@gmail.com (Y.-H.H.); 2Department of Electrical Engineering, National Chiayi University, Chiayi 600355, Taiwan; hongliang150413@gmail.com; 3Department of Electronic Engineering, National Taipei University of Technology, Taipei 106344, Taiwan; 4Department of Radiation Oncology, Fu Jen Catholic University Hospital, New Taipei City 24352, Taiwan; yujen.wang@gmail.com; 5School of Medicine, College of Medicine, Fu Jen Catholic University, New Taipei City 242062, Taiwan

**Keywords:** quality assurance, radiotherapy, stitching-based detector, pencil beam scanning, proton beam therapy, proton beam reconstruction, ionization chamber

## Abstract

In view of the great need for quality assurance in radiotherapy, this paper proposes a stitching-based detector (SBD) technique and a set of intelligent algorithms that can reconstruct the information of projected particle beams. The reconstructed information includes the intensity, sigma value, and location of the maximum intensity of the beam under test. To verify the effectiveness of the proposed technique and algorithms, this research study adopts the pencil beam scanning (PBS) form of proton beam therapy (PBT) as an example. Through the SBD technique, it is possible to utilize 128 × 128 ionization chambers, which constitute an ionization plate of 25.6 cm^2^, with an acceptable number of 4096 analog-to-digital converters (ADCs) and a resolution of 0.25 mm. Through simulation, the proposed SBD technique and intelligent algorithms are proven to exhibit satisfactory and practical performance. By using two kinds of maximum intensity definitions, sigma values ranging from 10 to 120, and two definitions in an erroneous case, the maximum error rate is found to be 3.95%, which is satisfactorily low. Through analysis, this research study discovers that most errors occur near the symmetrical and peripheral boundaries. Furthermore, lower sigma values tend to aggravate the error rate because the beam becomes more like an ideal particle, which leads to greater imprecision caused by symmetrical sensor structures as its sigma is reduced. However, because proton beams are normally not projected onto the border region of the sensed area, the error rate in practice can be expected to be even lower. Although this research study adopts PBS PBT as an example, the proposed SBD technique and intelligent algorithms are applicable to any type of particle beam reconstruction in the field of radiotherapy, as long as the particles under analysis follow a Gaussian distribution.

## 1. Introduction

According to various statistics from the literature, malignant tumors (cancer) have continually been listed among the top 10 causes of death worldwide [1,2,3]. Moreover, high mortality caused by cancer can be seen across various regions. For example, in Taiwan and South Korea, the dominant role played by malignant tumors remains evident [4,5]. To eradicate malignant tumors effectively, three types of therapies—surgery, chemotherapy, and radiotherapy (RT)—are the mainstream methods applied in modern medicine [6,7,8,9,10]. Among them, each possesses its own advantages and disadvantages and may be especially useful and appropriate in specific patient situations.

With several favorable characteristics, RT has gained increased attention and significance in malignant tumor treatment over the past few decades. On one hand, with the continual advancement of RT technology, medical doctors are increasingly finding RT indispensable in cancer treatment. On the other hand, surgery may encounter limitations in cases where the affected organs or surrounding tissues do not allow for surgical intervention. The number of such limiting conditions can be expected to increase based on recent statistics identifying a continuous rise in the number of malignant tumors occurring in the lungs, trachea, and prostate. Because the tissues in and around these organs are normally difficult to operate on, RT emerges as the preferred solution due to its non-contact nature. While eliminating the need to operate on patients, RT outperforms surgical therapy also in terms of patient recovery time. Finally, when a patient is too weak to undergo chemotherapy or surgery, RT offers a solution as a timely, powerful, and effective treatment method. The high popularity of RT is also evidenced by the fact that at least two-thirds of all cancer treatment regimens in the Western world now adopt this method. In short, RT is becoming increasingly important in modern medicine due to its essential and promising advantages over other cancer therapies. These major advantages include but are not limited to the clean procedure, non-invasive operation, effectiveness in eliminating tumors, and the precise and selective targeting of cancerous cells.

As an outstanding branch of cancer treatment, RT can be further classified into two subtypes—photon-based and particle-based therapy. Photon-based RT [11,12], also known as photon beam radiation therapy, uses photon beams, such as X-rays or gamma rays, to kill tumor cells. Particle-based RT [13,14], on the other hand, employs accelerated ions, such as helium or carbon ions, to eliminate malignant tumor cells. Of the two subtypes of RT, the particle-based one allows for improved control over the projected energy. Compared with its photon-based counterpart, particle-based RT is more capable of precisely concentrating the projected beam energy onto a certain depth of cells, thanks to the phenomenon of the Bragg peak. Given this unique advantage, particle-based RT enables more precise targeting without impacting the surrounding healthy cells. In addition, through superposition, Bragg peaks at different depths can be utilized to form a flat region where the energy can be maintained at a high and nearly constant level. In other words, the projected energy can be maintained over a wide range of tissue depths, instead of producing a peak at a single depth. In this way, the high-energy region resembles a plateau; this is a great improvement that can make the treatment process even more effective and efficient.

Of the ions used in particle-based RT, protons represent a unique type; they are formed by an ionized hydrogen atom [15]. Protons are unique because, compared with other types of ions, they have the lightest weight and the smallest physical size [16]. Based on their light weight, protons enable the highest controllability, since lighter ions facilitate the maneuvering of equipment, such as accelerators. In addition, thanks to their controllability and small size, protons enable the smallest area of attack, which helps to reduce the risk to neighboring healthy cells when the targeted tumor cells are eliminated. Because of these advantages, proton-based RT, also known as proton beam therapy (PBT), has gained prominence over other types of particle-based RT, which generally outperform their photon-based counterpart. Nevertheless, particle-based RT methods that adopt heavier ions, such as carbon, also have favorable attributes. For example, heavy ions can normally achieve higher relative biological effectiveness (RBE) because of their higher linear energy transfer (LET) [17].

As the medical equipment system continues to evolve, a state-of-the-art treatment technique, pencil-beam scanning (PBS) [18], was introduced to particle-based RT. The main idea of PBS is to accurately focus the particle beam energy on a small region. In this way, the scanning process can be performed in a pixel-by-pixel and layer-by-layer manner. In addition, the energy level of the particles in each projected pencil beam can be adjusted as needed. Such a combination of dose control flexibility and positioning accuracy boosts the high effectiveness of particle-based RT even further. Specifically, PBT that adopts the PBS algorithm is often referred to as pencil beam scanning proton therapy [19,20,21]. By integrating the two most advantageous methodologies, pencil beam scanning proton therapy has become one of the most potent and popular remedies in treating cancers today.

Recall that the projected protons are energized particles that are to be applied to human bodies. Therefore, despite the excellent performance of pencil beam scanning proton therapy in theory, it is still necessary to regularly monitor what is actually projected by medical equipment in practice. The main reason is to keep patients safe and to keep the treatment effective by making sure that quality assurance (QA) is satisfied [22,23,24,25,26]. For medical treatment, there are normally two types of QA definitions—machine-related QA and patient-related QA. Machine-related QA is about ascertaining whether a medical machine operates as expected quantitatively. Taking a PBT system as an example, whether the actual beam energy, the bombarded location, and the sigma value of the beam match the system’s expectation belongs to the field of machine-related QA. On the other hand, patient-related QA, leaning towards the human side, is concerned with the biological effects of a treatment on its recipients. Specifically, this research study focuses on machine-related QA and aims to develop an efficient algorithm to accurately reconstruct the information of the projected proton beam in pencil beam scanning proton therapy. To do so, the algorithm needs to retrieve three pieces of the most essential information concerning pencil beam scanning proton therapy from measurement data. The three pieces of information include the location, the sigma value, and the intensity of the projected proton beam that is under test.

Because of the importance of machine-related QA for pencil beam scanning proton therapy, scholars have been exploring various related issues [27,28]. A paper proposed in the *Electronics Journal* mentioned an optical fiber sensing detector for proton therapy. The method is interesting because the authors used the designed method to achieve a high spatial resolution result. However, the stitching architecture is very different from the paper in [29]. Hence, it may not be appropriate to make a direct comparison between our work and the paper in [29]. Another paper proposed at PTCOG 53 and supported by Pyramid company used a two-layer design to monitor the proton beam profile [30]. The alignment of the two layers is vertical; one is the coarse layer, and the other is the fine layer. By comparing the relative position, high spatial resolution can be yielded. Likewise, due to the fundamental difference between our work and [30], it is difficult to make a direct quantitative comparison. For detector system development, the recent work by Eichin shows great possibility [31]. By using the channel recycling technique, Eichin demonstrated a proton beam detection system with an active area of 18.75 cm in width and 26.25 cm in height. In addition, the pixel pitch of the detection system is 2.5 mm. Despite the great work by Eichin, this research study discovers that there is still room for improvement. Specifically, this research study notes that when a group of protons is projected, the places where they actually hit the target always follow a Gaussian distribution. By exploiting this observation, this research study develops and proposes a set of novel algorithms that can further enhance the resolution of the detection system for PBS proton therapy. By adopting the proposed algorithms, the corresponding active detection area is able to reach 25.6 × 25.6 cm^2^, which is larger than that of Eichin’s work by approximately 33%. In addition, the pixel resolution can be enhanced from the 2.5 mm in Eichin’s work to 0.25 mm in our research study. Compared with the spot sizes of current commercial PBT equipment, which normally range from 6 to 14 mm in radius, a 0.25 mm resolution is sufficiently high. The success of the proposed algorithms relies mainly on the long-tailed attribute of the Gaussian distribution. To accommodate the abrupt increase in the number of analog-to-digital converters (ADCs) caused by the new algorithms, this research study utilizes the technique of stitching-based detectors [32]. Through simulation, this research study verifies that the required number of ADCs can be successfully reduced to an acceptable level.

To present the proposed intelligent reconstruction algorithms and to demonstrate satisfactory and practical performance through simulation, this paper is organized as follows: Section 2 introduces the proposed algorithms. Section 3 demonstrates the simulation results and presents important findings. Section 4 discusses the achievements and possible limitations of this work. Section 5 concludes the achievements of this research study and suggests possible directions for future work.

## 2. Algorithm for Reconstructing Projected Proton Beam Information

To demonstrate the effectiveness of the proposed algorithms, this research study uses PBS proton therapy as an example. However, the idea of the proposed algorithms can be readily applied to any other particle-based RT as long as the particles used follow a Gaussian distribution. The sensing of protons starts from the electron gathering performed by a chunk of ionization chambers, the front-end sensors used for detecting electrons ionized by projected proton beams. Charge collection is primarily based on a multi-layered printed circuit board (PCB) ionization chamber structure, featuring four or more layers for efficient charge collection. The basic structure of this sensing module has been previously detailed in a Ph.D. dissertation [33] and a conference paper we proposed previously [32]. However, to avoid diverting the reader’s focus, a comprehensive description of these complex hardware details is omitted here. Furthermore, the proposed algorithms are not limited to a specific sensing instrument; they are designed to operate with parallel plate charge collection systems. While the relative direction between the electric field and the charged particle flow may influence charge collection efficiency, this potential issue can be effectively addressed through additional calibrations. For the whole ionization chamber plate, this research study assumes that it is square and composed of a total of 16,384 pixel-based ionization chambers distributed uniformly inside. That is, the constitutive ionization chambers inside the ionization chamber plate of our research study are placed in a 128 × 128 2D array. To represent the charge quantities sensed by these 128 × 128 ionization chambers, it is necessary to use a 128 × 128 2D array of real numbers, which is conceptually illustrated in Figure 1a. With a typical physical size of 2 × 2 mm^2^ for each ionization chamber, the size of our ionization chamber plate should be approximately 25.6 × 25.6 cm^2^.

As demonstrated in Figure 1a, the outermost square represents the 128 × 128 matrix that contains the information of charges collected by the 128 × 128 ionization chambers. Because of this reason, this research study addresses the outermost matrix in Figure 1a as the “physical layer,” which has a size of 128 × 128. Normally, to reconstruct the information of proton beams being projected, the data stored in the physical layer need to be further processed and analyzed. Since charges are inconvenient for data processing, it is conventional to transform the data in the physical layer into digital values by using ADCs. Without special attention, the number of ADCs should be the same as that of the ionization chambers, i.e., 16,384. However, 16,384 ADCs are not only expensive but also energy-consuming, two unfavorable attributes that hinder such naïve implementation. To reduce the number of ADCs to a practicable level, this research study utilizes a novel “stitching-based detector” technique together with an algorithm that assists the consequential operations. To explain how the proposed technique and algorithm work, this research study divides the physical layer in Figure 1a into 4 square groups, I, II, III, and IV. After division, each group, e.g., group I, becomes a 64 × 64 matrix. Then, this research study further divides each of the four groups into four areas, A, B, C, and D, each of which becomes a 32 × 32 matrix. Hence, each area, e.g., area A, contains 1024 components, which are numbered from 1 to 1024, as demonstrated in Figure 1b. The relationships between the physical layer, the four groups (I, II, III, and IV), and the four areas (A, B, C, and D), are depicted in Figure 1a.

To facilitate explanation, this research study hereafter abbreviates the “stitching-based detector” technique as the “SBD technique.” With the groups and sub-groups well defined, the SBD technique can be implemented by letting any four elements at the same relative location in their corresponding areas share one ADC. However, it is required that such ADC-sharing policy be applied only to areas with the same naming letter. For example, the four ionization chambers in the 1st row and 1st column within the area “A” of the four groups, “I,” “II,” “III,” and “IV”, should share the same ADC. This idea of ADC sharing is conceptually illustrated in Figure 2, where the ionization chambers in areas A, B, C, and D are marked with red, yellow, blue, and green. As can be seen in Figure 2, regions of the same color are connected together at their back ends. Such connections, however, are only for conceptual demonstration, indicating that this research study lets four elements satisfying the following conditions share a common ADC. The conditions are as follows: (1) the ionization chambers are at the same relative location within their corresponding areas, and (2) the areas should have the same color. By comparing Figure 1a and Figure 2, it is clear that the SBD technique can help reduce the number of required ADCs drastically by 75%, from 16,384 to 4096.

By applying the SBD technique, a 64 × 64 matrix suffices to store all the information of the 128 × 128 ionization chambers. For the sake of simplicity, this research study calls this 64 × 64 matrix the “data layer.” Although the SBD technique can help reduce the number of required ADCs to a reasonable level, the charge information of the four ionization chambers sharing a common ADC does become mixed. At first glance, this mixture of analog signals may seem to prevent the reconstruction of the projected proton beams under test. After further analysis, however, this research study discovers and proves a feasible method to untangle this seemingly insolvable problem. Through simulation, this research study verifies and concludes that the proton beams under test can be retrieved with high accuracy by exploiting the fact that projected protons must follow the Gaussian distribution.

Retrieving a proton beam under analysis means being able to derive three parts of information—the maximum intensity, the sigma value, and the location of the maximum intensity. Given the exorbitant cost of commercial proton-based radiative therapy, e.g., 1500 USD per hour, this research study will first prove the proposed algorithms through simulation. To simulate the information in the 128 × 128 physical layer in Figure 1a, this research study adopts an even larger matrix to serve as the data generator, a 1024 × 1024 matrix. Because this super-large matrix is only used to generate data and does not exist in the real world, this research study addresses it as the virtual layer for simplicity. In sum, three types of layers are being defined—the physical layer, the data layer, and the virtual layer. The relationships between these layers are depicted in Figure 3.

The original “unit module” uses a printed circuit board (PCB)-based ionization chamber (IC). In addition, the number of detection pixels of the IC is 32 × 32 = 1024. Each pixel is connected to a “real” wire, which is ultimately connected to the ADC port of electric chips, e.g., a TI DDC series current integrator. If we let each pixel be 2 mm × 2 mm and ignore the tiny spacing between pixels, the dimension of the “unit module” becomes 64 mm × 64 mm. Now, by making four copies of the “unit module,” each of which possesses 1024 wires, we can integrate the four “unit modules” as a square “integrated unit module.” Such an “integrated unit module” corresponds exactly to the data layer in Figure 3.

For simplicity, we address this “integrated unit module” as “IUM.” Clearly, the dimension of the IUM is 128 mm × 128 mm, and the total number of electric wires is 1024 × 4 = 4096. By focusing on the dimension of the IUM, it can be found that the sensing area is not wide enough for most clinical applications. When it is necessary to enlarge the sensing area to a size of 256 mm × 256 mm while keeping the spatial resolution unchanged, directly integrating four copies of the IUM may seem a straightforward and feasible solution. However, such an intuitive thinking leads to a potential problem, which is that a total of 4096 × 4 = 16,384 electric wires will become inevitable. Unfortunately, such a huge and complicated work, i.e., too many wires as well as ADC ports, makes the idea implausible and unattractive. It is to address this difficulty that this paper proposes the “stitching” concept and attempts to verify whether the proposed algorithms work or not.

To avoid the gigantic number of wires and ADC ports, the proposed algorithm does not make the “real” connections of four copies of the IUM. Instead, the algorithm lets the four copies of the IUM share the same set of wires as well as ADC ports, an economic arrangement that requires only 4096 wires in total. As is true with all algorithms in the world, the “reconstruction” of the proton beam also has its limitations. For example, the proposed design is suitable for the “pencil beam scanning” of a proton beam but less appropriate for a “wobbling proton beam.” Fortunately, pencil beam scanning is dominant in current advanced proton therapy. Hence, the proposed design is reasonable and practical. In addition, the spatial resolution can be achieved on a sub-millimeter scale by using the proposed numerical methods.

Making the virtual layer’s size 1024 × 1024 means that this research study uses an 8 × 8 matrix to simulate each element in the physical layer. Recall that each component in the physical layer stores the data sensed by the corresponding ionization chamber, which stands for one pixel. Hence, using an 8 × 8 matrix to simulate one pixel means that the resolution of a pixel can be enhanced by 64 times. The rationality of the setup for the virtual layer can also be accomplished by inspecting the Nyquist–Shannon sampling theorem. For the latest technology of ionization chamber manufacturing, one ionization chamber would occupy a physical area of 2 × 2 mm^2^. According to the Nyquist–Shannon sampling theorem, it takes at least twice the vertical and horizontal resolution to accurately reconstruct the proton beams’ information under analysis. This means that each element in the physical layer requires at least a 2 × 2 matrix to simulate in the virtual layer. To avoid possible failure in satisfying the Nyquist–Shannon sampling theorem caused by critical conditions, this research study utilizes an 8 × 8 matrix to simulate each element in the physical array. By making the size of the virtual layer 16 times larger than necessary, the resolution of the ionization chamber plate is enhanced from 2 mm to 0.25 mm.

Functionally, the proposed intelligent reconstruction algorithms aim to retrieve the three pieces of information concerning the proton beam under analysis based on the given information sensed by the 16,384 ionization chambers. Operationally, the derivation of the final answer, which comprises three main parts, is approached through a series of iterations, which are performed in different dimensions. Hence, to explain the whole algorithm set, the following paragraphs will take turns enunciating the partial iterations in each dimension.

A.Finding the maximum intensity

With increasing energy added to current proton beams, more and more ions can be generated in an ionization chamber. As the number of ions increases, it becomes more and more likely that the positive and negative ions will bump into each other and recombine. Such mechanism is called the ion recombination effect, which will reduce the charges detected by an ionization chamber and cause a further decrease in the current sensed by a back-end ADC. If the recombination effect is severe, in a situation where the proton beam intensity is high, the ADC is likely to pick up misleading information. That is, the ADC may sense a current level that is less than the real one. Therefore, to accurately retrieve the intensity of the proton beam under analysis, it is necessary to take the ion recombination effect into account. To do so, this research study adopts the famous Boag model [34] to reconstruct the current that ought to be sensed by an ADC if the ion recombination effect did not occur. The Boag model can be expressed by (1), where *I_ion_* is the current sensed by an ADC and *i_IC_* is the ionization current in an ionization chamber before any ion recombination happens. In addition, *A* is the area of an ionization chamber, *d* is the separation distance between the positive and negative plates in the ionization chamber, and *E* represents the external electric field. Finally, the number 673 ± 0.8 in (1) is a constant presented in [35].(1)$$I_{ion}=i_{IC}\times \frac{1}{\left(1+(673\pm 0.8)\times E^{-2}\times \frac{i_{IC}}{A}\times d^{2}\right)}$$

By using (1), the value of *i_IC_* can be derived from *I_ion_* when all the other variables are known. For example, for a proton beam projected onto the center of the physical layer, it is feasible to focus only on the ionization chambers located on the middle-positioned y-axis. Under this assumption, Figure 4 illustrates the magnitudes of the sensed current, specified on the y-axis, with respect to the 1024 elements in the virtual layer, specified on the x-axis. The unit of the current magnitude in Figure 4 is “ADC count,” which equals a current level of 0.1 pA. Figure 4a and 4b demonstrate the conditions when the maximum sensed current levels are 10^6^ ADC and 10^4^ ADC, respectively. The blue and red curves in Figure 4 represent the real and ideal current levels sensed with and without the ion recombination effect. By comparing Figure 4a,b, it is noticeable that the ion recombination effect is more severe when the proton beam has higher intensity. To execute the Boag model correctly, several important inner parameters of the ionization chambers need to be known. Table 1 lists the values of these parameters for this research study.

The procedures for deriving the maximum intensity of the proton beam under analysis are depicted in Figure 5, where the “I_0_” represents the maximum current found in the data layer. By using (1), the maximum current that should be sensed without the ionization effect can be computed. This compensated current level will be fed into the virtual layer, which, serving as the data generator, will generate the corresponding proton beam information. The generated data will then be simulated to account for the ion recombination effect before being sent to the physical layer to simulate charge detection and to the data layer to simulate channel combination. Finally, the new maximum current level, “I_pred_,” will be compared with the old maximum current level, I_0_. If the absolute value of the difference between I_0_ and I_pred_ is larger than the threshold, i.e., 0.01%, the algorithm will continue to iterate the same process. On the other hand, if the absolute value of the difference between I_0_ and I_pred_ is less than the threshold, the algorithm will stop and report the derived maximum current level that results in I_pred_.

B.Finding the initial estimation of the sigma value

This research study uses two phases to search for the sigma value of the proton beam under analysis. The first phase is to make a preliminary guess about the sigma value by exploiting a special feature of the Gaussian distribution. That is, the fact that the full width at half maximum (FWHM) is 2.355 times the sigma value. Since any projected proton beam follows the Gaussian distribution, this research study uses this property to acquire the initial estimation for the sigma value. Figure 6a illustrates the determination of the estimated sigma, whereas Figure 6b shows the corresponding algorithm. As shown in Figure 6b, the algorithm first searches the 64 × 64 data layer and finds all the elements with intensity equal to or higher than half the maximum current, “I_0_.” Then, the algorithm uses the element with the maximum intensity, I_0_, as the center point, and counts the number of yellow elements in the upward, downward, leftward, and rightward directions. By choosing the maximum among the numbers of satisfied elements in the four directions, the algorithm can derive and output a decent preliminary guess for the sigma of the proton beam under analysis. As an example, the left side of Figure 6 illustrates the center point, and the right side of Figure 6 marks the satisfied elements with yellow. Using Figure 6a as an example, the numbers of satisfied elements in the upward, downward, leftward, and rightward directions with respect to the center point are 10, 9, 9, and 10, respectively. This means that the initial estimation of the sigma value should be 10 pixels.

C.Finalizing the sigma value

Influenced by the ion recombination effect, the preliminary estimation of the sigma value still possesses a certain error. Hence, this research study attempts to finalize the sigma value at this stage of operation. The proposed algorithm is explained in Figure 7. Initially, the algorithm uses the output in Figure 5 as the maximum intensity, the location of the maximum intensity as the center, and the output in Figure 6b as the sigma value. By feeding these combinations of information into the virtual layer, a corresponding formation of the data layer can be obtained. Then, the algorithm compares the formation of the newly generated data layer with that of the old one to decide the adjustment of the sigma value. In addition, this research study finds that normalizing the maximum intensity before performing the comparison can facilitate the derivation of the optimal sigma value. Hence, a standardization step is placed before the comparison step, an insertion that can be noticed in Figure 7. After the normalization, the algorithm sums up all the intensities of the 8 × 8 elements with the pre-defined center located in the middle. Such summation is executed on both the newly generated data layer and the original one. If the summation of the newly generated data layer is larger than that of the original one, the algorithm subtracts the guessed sigma value by one. On the other hand, if the summation of the newly generated data layer is less than that of the original one, the algorithm adds the guessed sigma value by one. The same procedure is performed until an acceptable error level is met, where an output of the sigma value is reported.

D.Finalizing the location of the maximum intensity in the physical layer

Up till now, the proposed algorithms only analyze the information gathered from the data layer. This means that the actual location of the maximum intensity on the physical layer has not been ascertained yet. Hence, the final step of proton beam reconstruction is to decide the location of the maximum intensity on the physical layer.

Recall that the information in the data layer is acquired by summing up that of the four quadrants in the physical layer. This means that it is unknown which quadrant contains the location with the maximum intensity and therefore contributes the strongest signal. To solve this conundrum, this research study proposes a final algorithm. The algorithm first maps the location with the maximum intensity in the data layer to the four corresponding points in the four quadrants in the physical layer. The mapping is illustrated in Figure 8. Moreover, to correct the possible error in deciding the location with the maximum intensity, the algorithm extends the searching area from the original center outward to nine elements with the original center in the middle. The extension is demonstrated by the 3 × 3 gray matrix in Figure 8. This means that a total of nine elements become potential candidates for being the location with the maximum intensity for each quadrant. Since the physical layer contains four quadrants that sum up to form the data layer, the algorithm will consider a total of thirty-six elements in the physical layer as candidates for the location with maximum intensity.

To make the best guess about the real location of the maximum intensity in the physical layer, the algorithm integrates the output from Figure 5 and Figure 7 with the thirty-six elements illustrated in Figure 8. By feeding these combinations of information into the virtual layer, a total of thirty-six kinds of data layer formations can be generated. Finally, by comparing the thirty-six kinds of data layer formations with the original data layer, the algorithm is able to select the one with the lowest error.

The overall proton beam information reconstruction algorithm, from subsection “A” to subsection “D,” is summarized in Figure 9, where the maximum intensity is first derived. The second step is to estimate the initial value of the sigma value, followed by the third step, which aims to improve the sigma estimation. Finally, with every piece of information in hand, the location with the maximum intensity in the physical layer can be identified at the fourth step.

Knowing that the data layer is formed by applying the SBD technique to the physical layer, one may feel puzzled as to how the proposed algorithms can work successfully. The reason simply lies in the fact that protons in any proton beam need to follow the Gaussian distribution, which is exploited by this research study. To explain the reason more vividly, Figure 10 illustrates four extreme conditions where four proton beams are projected onto the centers of the four quadrants of the data layer. The orange color areas represent the places with the highest intensity, whereas the green color areas represent the places with the lowest intensity. At first glance, it may be tempting to think that the data sensed by the ADCs for the four scenarios should be identical. This misconception results mainly from two observations. First, areas “A,” “B,” “C,” and “D” in the four sub-matrices share the same sets of ADCs. Second, the four proton beams are projected onto the intersection of A, B, C, and D in the four quadrants. Despite the misleading first-glance impression, the simulation results presented through the images in the right column of Figure 10 show that the data sensed by the 64 × 64 ADCs demonstrate different behaviors. For example, the first condition has a conspicuous difference from the second at the top right part of the sensing area. The major difference between the first and the third conditions, on the other hand, happens at the top part of the sensing area. And the most significant difference between the first and the fourth conditions also lies in the top right part of the sensing area. Interestingly, in spite of the same location where major differences occur, the difference between the first and the second conditions and that between the first and the fourth conditions are still different. As a matter of fact, the four images in the right column of Figure 10 are all different and unique. In other words, the simulation results have proven that the proton beam projection in the four extreme scenarios still produces sufficient differences in the data sensed by the data layer. It is by exploiting these differences that the proposed algorithms implement the inverse function and reconstruct the original proton beam under test.

## 3. Simulation Results and Important Findings

To verify the effectiveness of the proposed SBD technique and reconstruction algorithms, this research study adopts two levels as the maximum intensity detected in the physical layer. The two levels are 10^4^ ADC and 10^6^ ADC, respectively, where 1 ADC equals a current level of 0.1 pA. Table 2 shows the simulation results when the maximum intensity is set to 10^4^ ADC and the sigma value is changed from 10 to 120 with a step of 10. Furthermore, Table 2 regards any predicted location of the maximum intensity within +/− 2 pixels from the real location, equal to +/− 0.5 mm, as a correct reconstruction. For each sigma value, this research study randomly chooses an element from the physical layer as the location with the maximum intensity and runs the reconstruction simulation. By executing this process 2000 times for each sigma, the corresponding error rate of the proposed algorithms with respect to different sigma values can be computed. By extending the definition of a correct reconstruction to +/− 4 pixels, Table 3 shows that the error rates become even smaller, a phenomenon proving the great performance of the proposed algorithms. By integrating the information in Table 2 and Table 3, Figure 11 reveals the overall error rates with respect to different sigma values and definitions of a correct reconstruction.

By adopting all the other settings and definitions in Table 2, Table 4 shows the simulation results when the maximum intensity is set to 10^6^ ADC. Likewise, Table 5 shows the simulation results when the maximum intensity is set to 10^6^ ADC by adopting all the other settings and definitions in Table 3. Despite some changes in the error rates under different circumstances, the error rates are still well below 5%, a good phenomenon demonstrating the robustness of the proposed algorithms. By combining the information in Table 4 and Table 5, Figure 12 displays the overall error rates with respect to different settings and definitions with intensity equal to 10^6^ ADC.

To evaluate the level of randomness in our simulation, this research study particularly counts the number of cases where the centers of proton beams lie near the boarders of A, B, C, and D. Specifically, this research study defines regions where the x coordinate is less than 15, between 245 and 265, between 757 and 777, and larger than 1008 in the virtual layer as being close to the boundary. And the same definition is applied to the y coordinate. The defined regions are marked in blue in Figure 13. By using the above definition, the number of proton beams with respect to different sigma values and maximum intensities is shown in Figure 14. For each combination of sigma value and maximum intensity, a total of 2000 proton beams are generated for simulation. As can be observed, regardless of the sigma values and maximum intensities, the number of satisfied proton beams is approximately constant. In addition, the ratio of such constant to 2000 is approximately that of the defined area to the size of the virtual layer, a phenomenon confirming the randomness of the simulation.

To visually examine the erroneous cases in Table 2, Table 3, Table 4 and Table 5, Figure 15, Figure 16, Figure 17, Figure 18, Figure 19, Figure 20, Figure 21 and Figure 22 show their corresponding locations in the virtual layer. Specifically, Figure 15 and Figure 16 illustrate the erroneous cases in Table 2 with sigma values varying from 10 to 60 and from 70 to 120, respectively. Likewise, Figure 17 and Figure 18 display the erroneous cases in Table 3 with sigma values varying from 10 to 60 and from 70 to 120. In the same fashion, Figure 19 and Figure 20 display the erroneous cases in Table 4, whereas Figure 21 and Figure 22 demonstrate the erroneous cases in Table 5. From Figure 15, it can be seen that most errors happen near the A-B-C-D boundary as well as the outer boundary of the virtual layer. This phenomenon stands to reason because these locations are more likely to cause misjudgment due to the innate attribute of the proposed SBD technique. By comparing Figure 15 and Figure 16, it is observable that all the erroneous cases near the A-B-C-D boundary disappear with sigma values greater than 60. That is, only the errors occurring near the perimeter of the virtual layer remain with larger sigma values. This indicates that the proposed SBD technique and reconstruction algorithms have better performance with larger sigma values. Such a finding is logical because proton beams with smaller sigma values tend to suffer more imprecision near the boundaries of symmetry, such as the A-B-C-D boundary and the peripheral boundary. It should be remembered, however, that proton beams are not aimed near the outer boundary of the virtual and physical layers in real practice. This means that the errors in Figure 16 can be practically regarded as nonexistent in real applications of pencil beam scanning PBT. By comparing Figure 15 with Figure 17 and Figure 16 with Figure 18, it can be noticed that the errors are slightly reduced. This is simply because Figure 17 and Figure 18 extend the definition on an erroneous case from +/− 2 pixels to +/− 4 pixels. By comparing Figure 15 with Figure 19 and Figure 16 with Figure 20, it can be seen that there is no significant change, except for an eye-catching error inside the virtual layer in Figure 20. Furthermore, this unique error, an error not near the boundary of the virtual layer with sigma values larger than 60, still remains when the maximum intensity is changed to 10^6^ ADC in Figure 22. Although the actual cause of this particular error is still under investigation, the problem is very likely to result from the poor numeric precision provided by the Python programming language (version 3.11.5) utilized by this research study. Finally, in Figure 19 and Figure 20 and in Figure 21 and Figure 22, there are no obvious differences, except for the smaller number of errors due to the layer definition of an erroneous case.

To further verify the proposed algorithms’ validation, the relative errors of predicted amplitudes with different variances of proton Gaussian beams are computed and shown in Table 6 and Table 7.

Basically, the full width at half maximum (FWHM) is approximately two times the variance of the Gaussian profile, and the FWHM is usually within 10 mm–30 mm in most cases clinically. Table 6 and Table 7 show the variance of the proton Gaussian beams within 20–60 pixels in the virtual layer scale that corresponds to 20 × 2 × 0.25 mm—60 × 2 × 0.25 mm scales. The results fit the FWHM of PBS beams properly. Moreover, the results show that the amplitude of the broad Gaussian beam could be estimated well, e.g., sigma = 120, which means FWHM = 60 mm. The simulation indicated that the relative error of the amplitude reconstruction is less than 3% only with the super-narrow Gaussian proton beam, e.g., variance = 10 in Table 6 and Table 7. Therefore, the simulation results are satisfactory and practical.

## 4. Discussion

This research study proposes the novel SBD technique and intelligent reconstruction algorithms to realize high spatial resolution of dosimetry in pencil beam scanning proton therapy. Although the pixel size of the ionization chamber adopted by this research study is 2 × 2 mm^2^, the effective resolution can be substantially enhanced to 0.25 mm by using the proposed intelligent reconstruction algorithms. Compared with current detectors on the market, the SBD technique demonstrates excellent performance in spatial resolution. For example, the 7.1 mm center-to-center distance between isolated ionization chambers of OCTAVIUS Detector 1500XDR and the 2.5 mm spatial resolution in the central area of OCTAVIUS Detector 1600XDR are both inferior to the 0.25 mm resolution achieved by this research study. In addition, by exploiting channel sharing, the proposed stitching arrangement successfully reduces the number of ADCs from 16,384 to 4096. Such a significant reduction of 75% in ADCs greatly saves hardware implementation costs and power consumption. Moreover, the proposed architecture is able to support precise reconstruction even for large cross-regional proton spots, e.g., with σ equal to 120 and a physical diameter approximately equal to 70.6 mm. In fact, the proposed architecture can support a maximum diameter of 128 mm for proton spot reconstruction. Compared with related works [31], these advantages show that the proposed method offers much better practicality and expandability.

For the data processing aspect, the simulation results reveal that it takes about five to eight minutes to reconstruct the data for a proton spot. Such processing time includes Boag model compensation, σ estimation, and other complicated algorithms, such as the reconstruction of the maximum intensity location. The hardware used for simulation was a 12th Gen Intel(R) Core(TM) i7-12700 processor, which operates at 2.1 GHz. During the simulation, this research study swept the σ values from 10 to 120 and randomly generated 2000 data points for each σ value. Hence, a total of 24,000 data points were simulated for the reconstruction test. Although the present hardware environment was sufficient to support the amount of computation, it is a good idea to replant the whole system on a GPU-based platform in the future. Using a GPU to process data facilitates immediate response by reducing the computation time by 1/10 or 1/20, an improvement that may enable the overall processing time to be within only a few minutes. The proposed SBD technique and intelligent reconstruction algorithms aim at pencil beam scanning proton therapy, which is characterized by concentrating proton beams on small areas using pixel-to-pixel and layer-to-layer scanning. Hence, the main goal of the proposed algorithms is to reconstruct the information of a single proton spot, including the intensity, the σ value, and the location of the maximum intensity. This means that the proposed algorithms can effectively handle common QA scenarios involving single-point measurements, e.g., grid patterns of a 7 × 7 array within a 25 cm × 25 cm field, assuming that the spots in the array are composed of independent proton spots.

Nevertheless, this research study also has its limitations and may need to be improved in the future. The maximum error rate found by this research study, 3.95%, is derived by pure numerical simulation without utilizing any post-processing calibration procedures that may be applied in real-world clinical applications. According to the IAEA TRS-398 standard [35], the maximum error rate (Maximum Relative Deviation) needs to be less than 1% to be considered acceptable because even minor residual errors may have a clinical impact in high-precision radiotherapy. Therefore, the proposed system will need to be further integrated with clinical calibration procedures for future clinical implementation.

In addition to the calibration issue, the proposed stitching technique also faces a challenge of unequal gains. Different lengths of signal transmission paths are likely to cause different parasitic resistances and capacitances. Although the influence of parasitic capacitances may be neglected due to the low operating frequency of the system, the gain is still affected by different parasitic resistances. Such a difference in gain needs to be carefully adjusted to guarantee the accuracy and conformity of dosimetry. Due to space limitations in the paper, details about the calibration and adjustment of the hardware implementation will be further presented in the next paper.

Within the current implementation framework, the particle beams are assumed to follow an ideal symmetric Gaussian distribution, based on which the reconstruction can be accomplished. Such a simple assumption helps simplify the mathematical models and evaluate whether a super-high-resolution reconstruction can be implemented when the number of read-out channels is deliberately reduced. Although beam symmetry in clinical settings may vary depending on the manufacturer and system configuration (e.g., magnet arrangement, ion source stability, and beamline design), vendors would normally strive to optimize the pencil beam into a highly symmetric, narrow Gaussian distribution to ensure the precision of dose delivery. If the actual beam profile deviates significantly from the Gaussian model, the reconstruction results derived through the algorithms may become inaccurate. Under these circumstances, it may be necessary to adopt more advanced models, such as convolution-based models or machine learning methods, to correct the errors. In other words, future practical applications may require individual calibration and adjustment based on specific beam characteristics and circuit gain variations. Furthermore, the present algorithms cannot process uniform square fields (e.g., 10×10 cm). For example, the proposed stitching PCB-based ionization chamber will fail in wobbling proton therapy because the wide beam uses magnetic fields and a ridge filter operation. This limitation arises because our algorithms are specifically designed for pencil beam scanning proton therapy, which possesses the narrow-Gaussian-profile characteristic. On the other hand, the proton distribution in uniform square fields differs substantially from the Gaussian assumption. Hence, the proposed algorithms cannot be directly applied to uniform square fields.

Finally, ion recombination effects in ionization chambers are strongly influenced by the dose rate of the proton beam. Under high-dose-rate conditions, elevated ion densities in the chamber can lead to partial charge loss due to recombination, a mechanism that can compromise the accuracy of dose readings. As this study is simulation-oriented, it adopts idealized dose rate conditions based on vendor specifications. Hence, the effects of ion recombination are not further addressed in this research study. In future clinical applications, however, considering the ion recombination effects in high-dose-rate conditions may be necessary, and appropriate adjustments may be required according to the real situations.

In sum, this study is currently in an early simulation stage, where the main purpose is to verify whether the proposed theorems may work. Based on the preliminary success in realizing ultra-resolution reconstruction and in reducing the number of required ADCs, this research study has laid a solid theoretical foundation for the future development of high-resolution proton beam dosimetry techniques. As for the unresolved issues, such as numerical errors, non-uniform circuit gain, beam symmetry assumptions, and ion recombination, they will be properly addressed and introduced in future work and forthcoming publications.

## 5. Conclusions

This research study proposes an SBD technique and a set of intelligent algorithms to reconstruct a proton beam under test. The SBD technique and intelligent algorithms are verified through simulation, where pencil beam scanning PBT is adopted as the scenario under assumption. The simulation results show that the novel idea of our work has a great performance and a very low error rate in reconstructing the information of the proton beams under analysis. The simulation adopts two maximum intensity levels, which are 10^4^ ADC and 10^6^ ADC. With a 10^4^-ADC maximum intensity and sigma values between 10 and 60, most errors happen near the A-B-C-D boundary and the perimeter of the virtual layer. When the sigma values are changed between 70 and 120, errors happen only near the perimeter of the virtual layer. A similar trend can be noticed by changing the maximum intensity to 10^6^ ADC, except for one particular error not near the perimeter of the virtual layer. Although the real cause of this unique error is still under investigation, it is highly suspected to result from the poor numeric precision provided by the Python language. Nevertheless, the error rates in all simulated scenarios are stable and extremely low, a good phenomenon reconfirming the effectiveness and efficiency of the proposed technique and algorithms. The maximum error rate among all the simulated scenarios is 3.95% when the sigma value is 10, the maximum intensity is 10^6^ ADC, and the error definition is +/− 2 pixels. In addition, since proton beams are normally not aimed near the periphery of a sensor area, the errors happening near the periphery of the virtual layer can be ignored. This means the actual error rate of the proposed method can be even lower when used in real applications. Table 6 and Table 7 reconfirm the practicality and accuracy of the proposed method. Simulation results show that the method can accurately reconstruct the amplitude of both wide proton beams, i.e., with FWHM = 60 mm, and extremely narrow proton beams with a relative error below 3%. This indicates that the simulation method possesses robust stability and practical applicability. Moreover, although PSB PBT is used for demonstration, the proposed method can be readily applied to any other particle beams under test as long as the constitutive particles follow the Gaussian distribution.

## Figures and Tables

**Figure 1 sensors-25-04985-f001:**
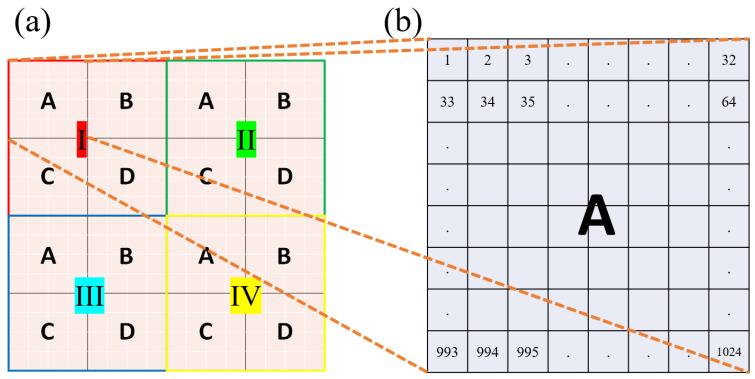
(**a**) The physical layer is a 128 × 128 matrix storing the charge quantities collected by all the 128 × 128 ionization chambers. The groups I, II, III, and IV are four sub-matrices of the physical layer. Each group is further divided into four sub-matrices, A, B, C, and D. (**b**) Each area is a 32 × 32 matrix, whose elements can be numbered from 1 to 1024 in left-to-right and top-to-bottom order.

**Figure 2 sensors-25-04985-f002:**
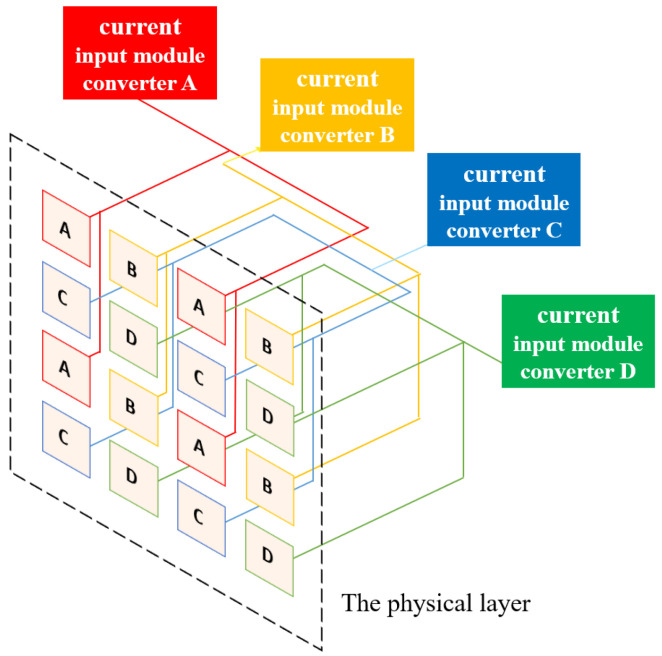
A conceptual illustration of the proposed SBD technique.

**Figure 3 sensors-25-04985-f003:**
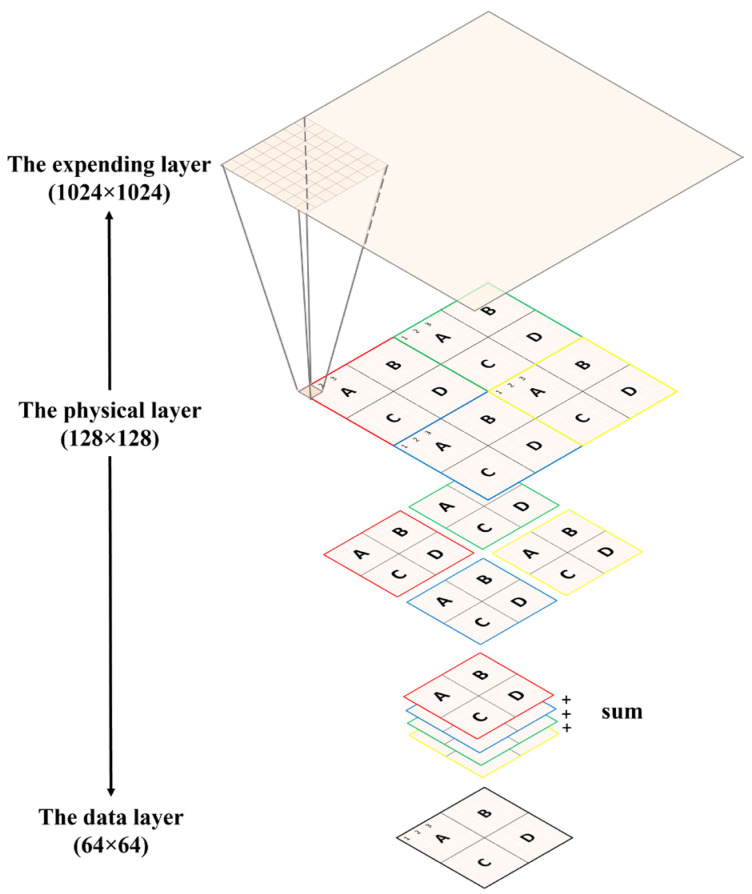
The relationships between the virtual layer, physical layer, and data layer. Each pixel in the physical layer is an 8 × 8 matrix in the virtual layer. Thus, the virtual layer can be used to reconstruct the ultra-high-resolution image.

**Figure 4 sensors-25-04985-f004:**
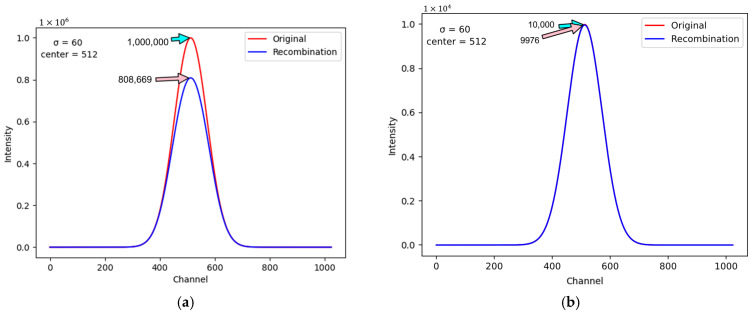
(**a**) The ion recombination effect is more severe with higher proton beam intensity. (**b**) The ion recombination effect is less obvious with lower proton beam intensity.

**Figure 5 sensors-25-04985-f005:**
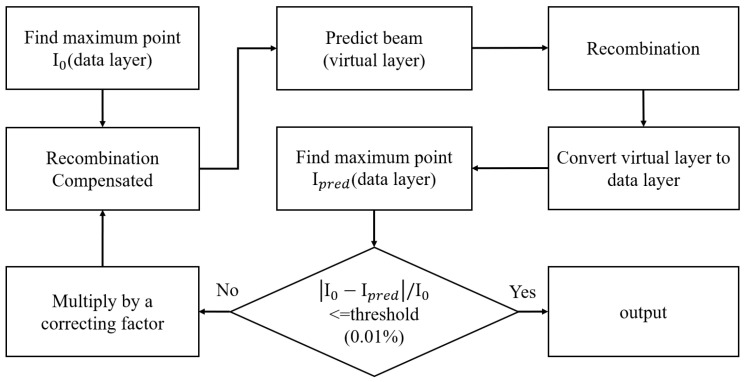
The procedures for deriving the maximum intensity of the proton beam under analysis.

**Figure 6 sensors-25-04985-f006:**
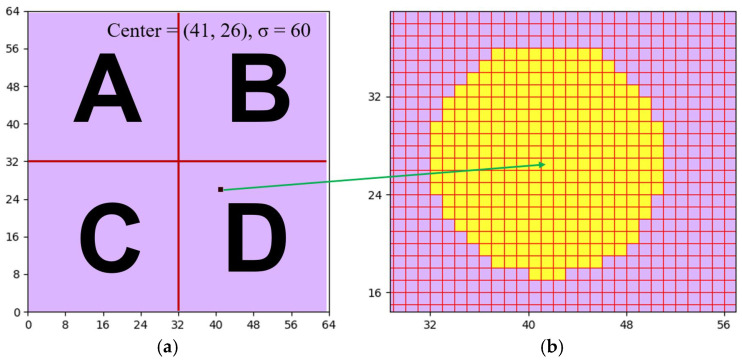
(**a**) The located spot with the maximum intensity in the data layer, with A, B, C, and D corresponding to the A, B, C, and D areas in Figure 3. (**b**) All the spots that possess an intensity higher than 50% of the maximum intensity.

**Figure 7 sensors-25-04985-f007:**
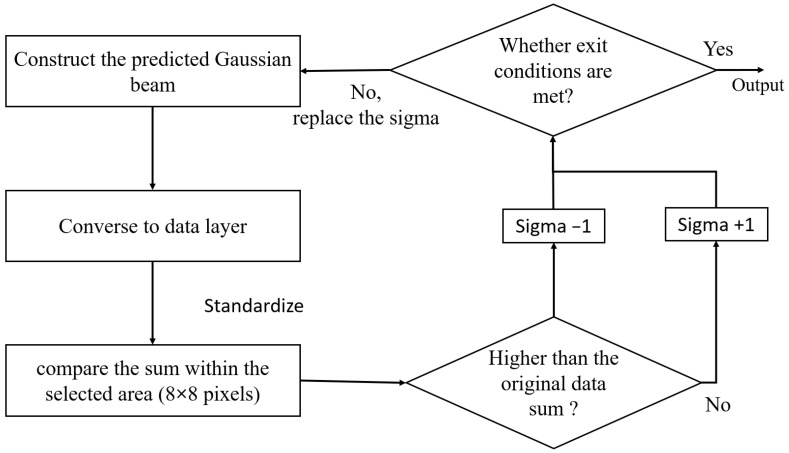
The algorithm for finalizing the location of maximum intensity and the sigma value.

**Figure 8 sensors-25-04985-f008:**
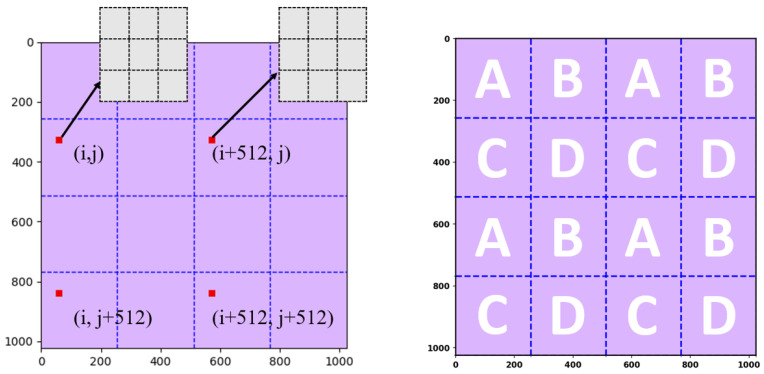
Mapping the location with the maximum intensity in the data layer to the four possible points in the physical layer.

**Figure 9 sensors-25-04985-f009:**
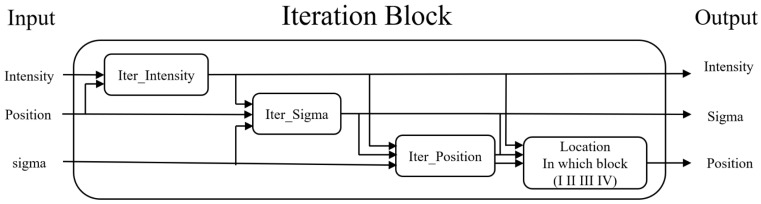
The proposed algorithms can reconstruct the location of the projected proton beam center.

**Figure 10 sensors-25-04985-f010:**
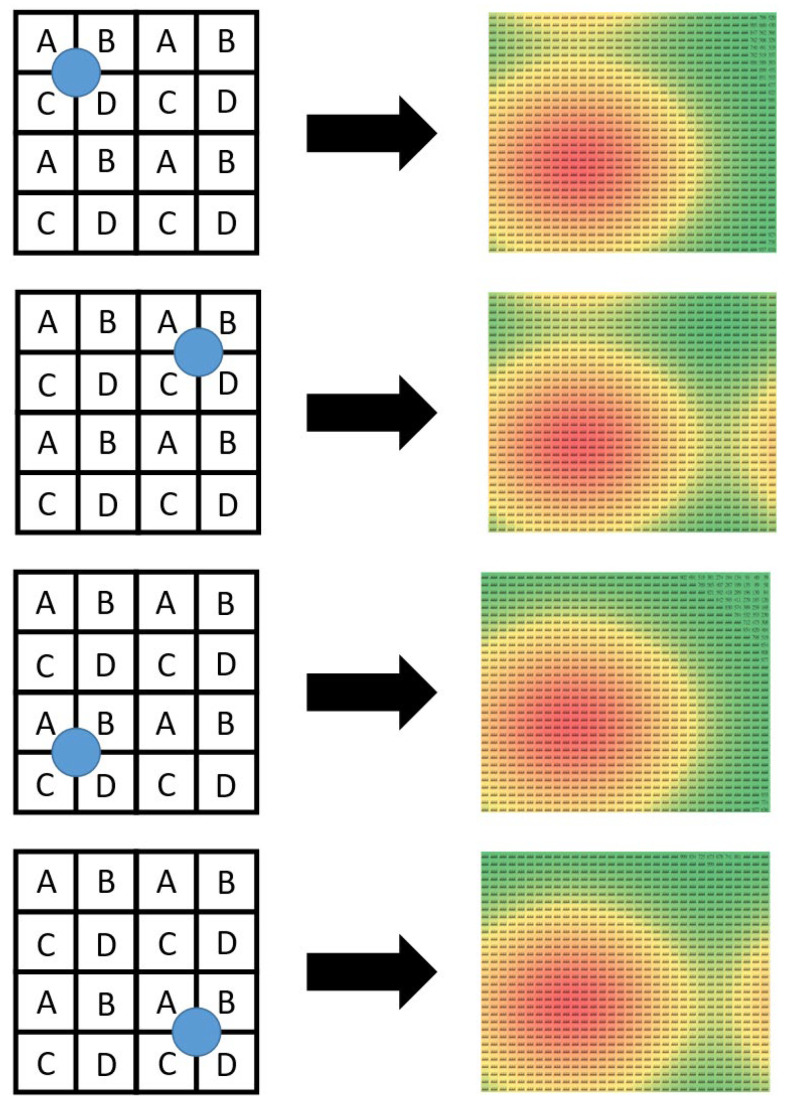
The information collected by the 64 × 64 data layer is sufficient for reconstructing the location of the proton beam center.

**Figure 11 sensors-25-04985-f011:**
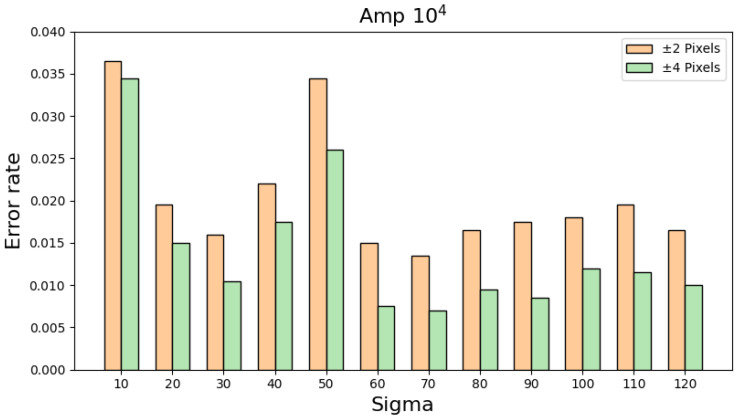
A visual illustration of the information in Table 2 and Table 3.

**Figure 12 sensors-25-04985-f012:**
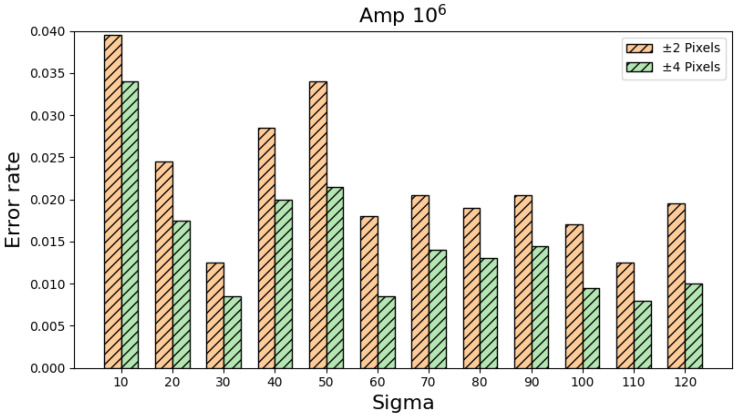
A visual illustration of the information in Table 4 and Table 5.

**Figure 13 sensors-25-04985-f013:**
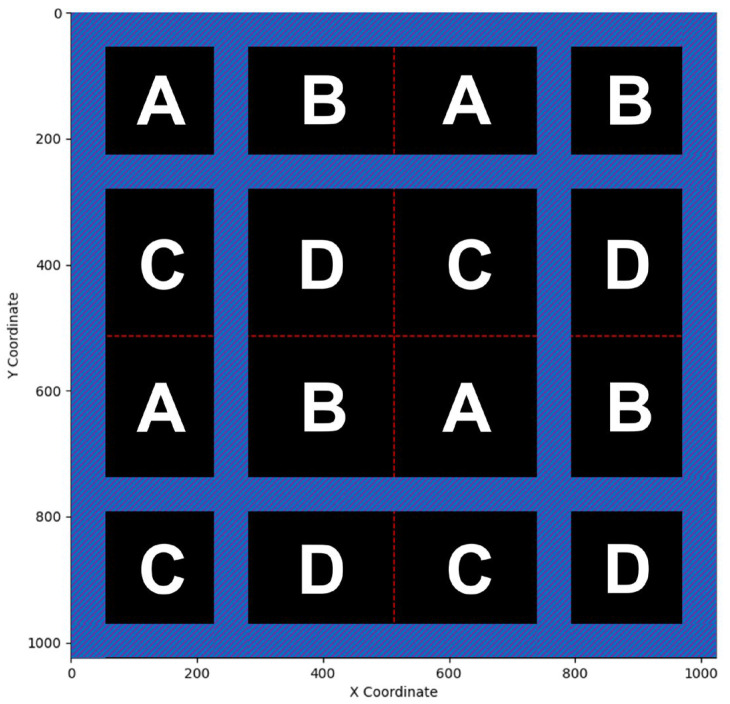
The blue regions are the areas defined as being close to the borders.

**Figure 14 sensors-25-04985-f014:**
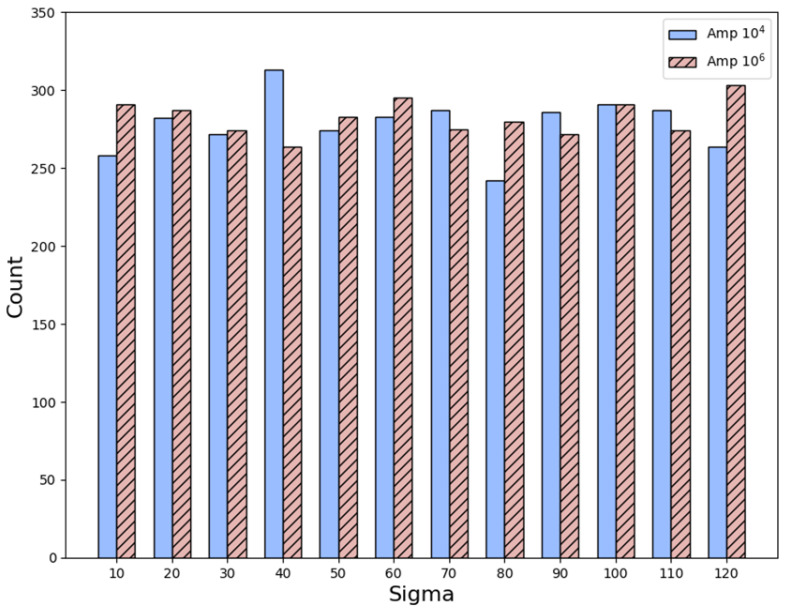
The number of randomly generated proton beam centers that fall within the boundary region for different sigma values and maximum intensities.

**Figure 15 sensors-25-04985-f015:**
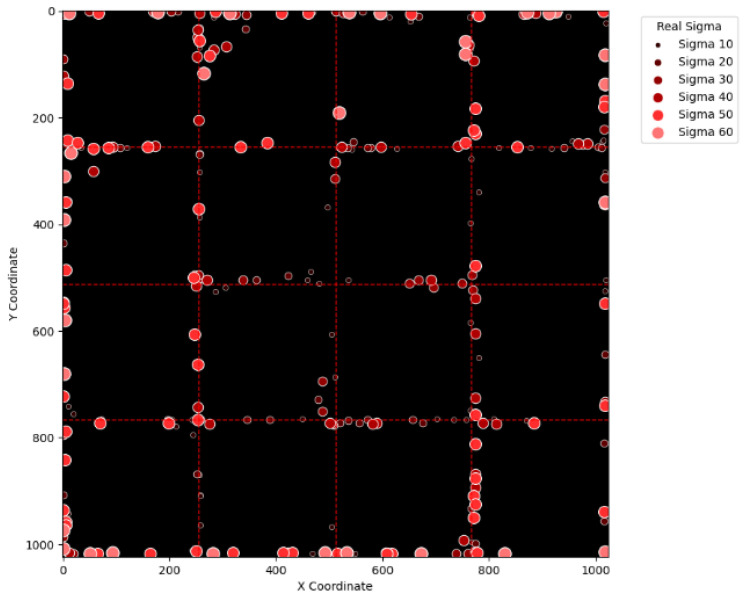
The locations of erroneous cases corresponding to the sigma values varying from 10 to 60 in Table 2. (Maximum intensity = $10^{4}$ ADC and error ≡ +/−2 pixels.)

**Figure 16 sensors-25-04985-f016:**
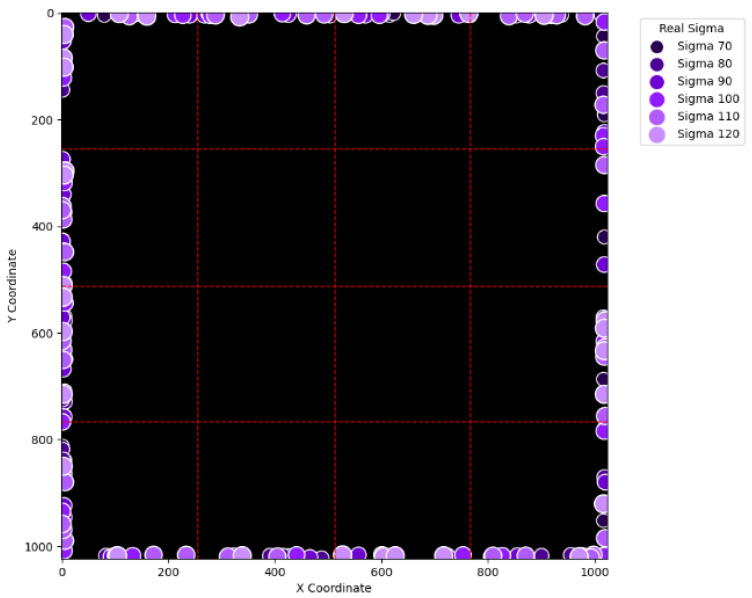
The locations of erroneous cases corresponding to the sigma values varying from 70 to 120 in Table 2. (Maximum intensity = $10^{4}$ ADC and error ≡ +/−2 pixels.)

**Figure 17 sensors-25-04985-f017:**
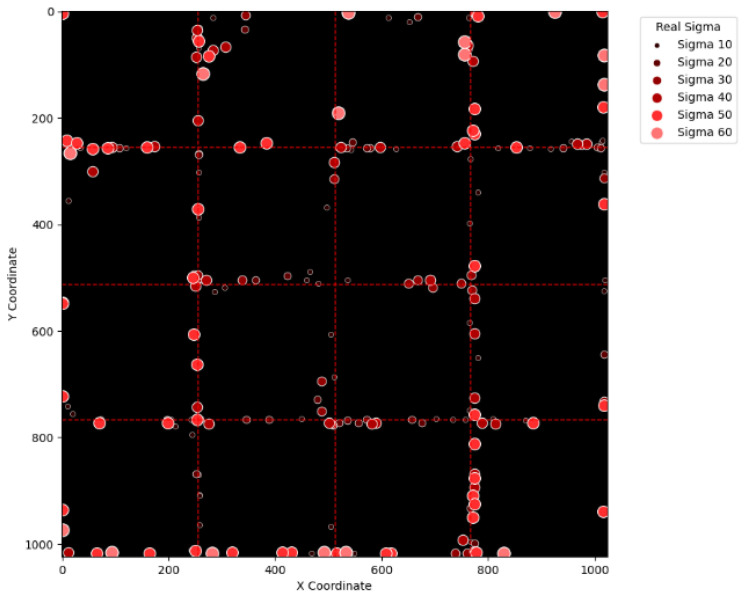
The locations of erroneous cases corresponding to the sigma values varying from 10 to 60 in Table 3. (Maximum intensity = $10^{4}$ ADC and error ≡ +/−4 pixels.)

**Figure 18 sensors-25-04985-f018:**
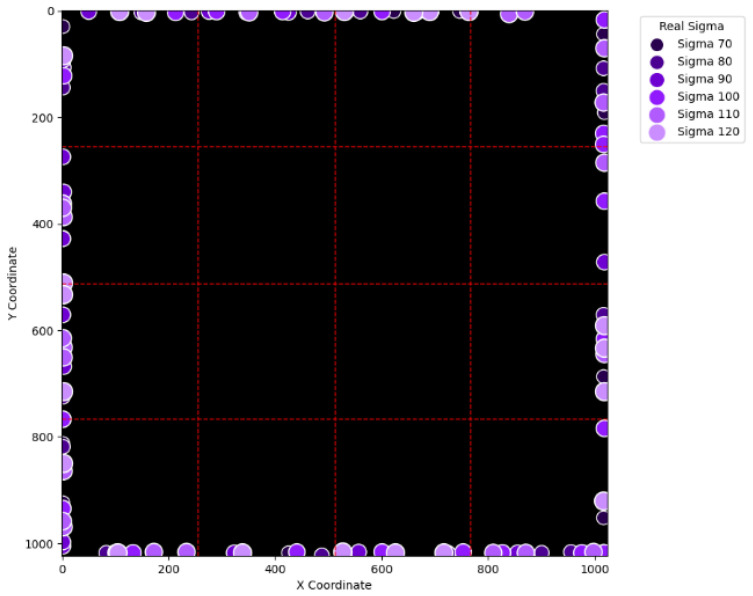
The locations of erroneous cases corresponding to the sigma values varying from 70 to 120 in Table 3. (Maximum intensity = $10^{4}$ ADC and error ≡ +/−4 pixels.)

**Figure 19 sensors-25-04985-f019:**
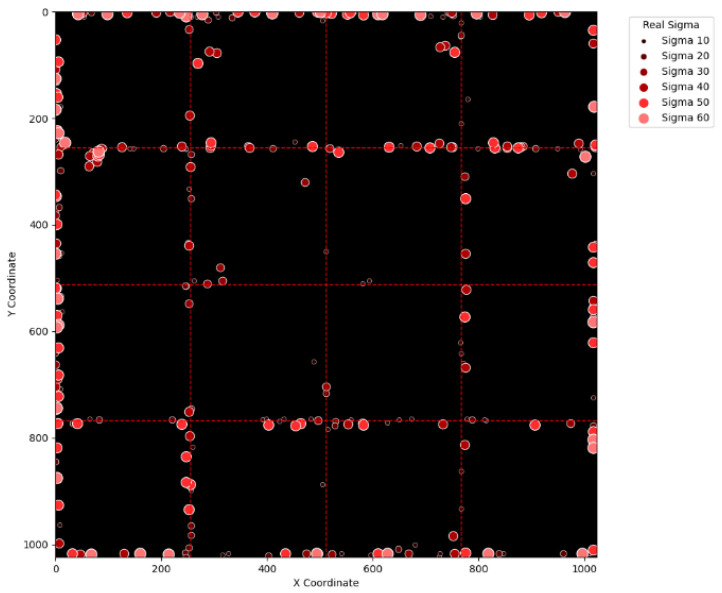
The locations of erroneous cases corresponding to the sigma values varying from 10 to 60 in Table 4. (Maximum intensity = $10^{6}$ ADC and error ≡ +/−2 pixels.)

**Figure 20 sensors-25-04985-f020:**
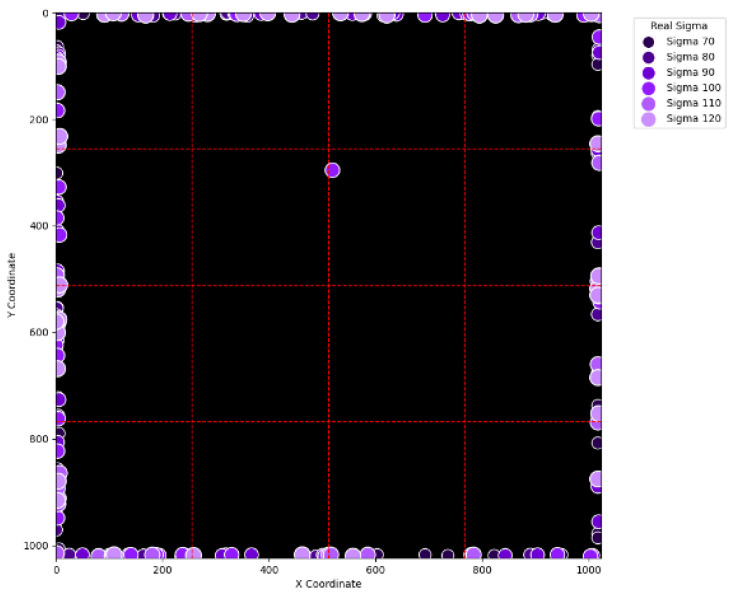
The locations of erroneous cases corresponding to the sigma values varying from 70 to 120 in Table 4. (Maximum intensity = $10^{6}$ ADC and error ≡ +/−2 pixels.)

**Figure 21 sensors-25-04985-f021:**
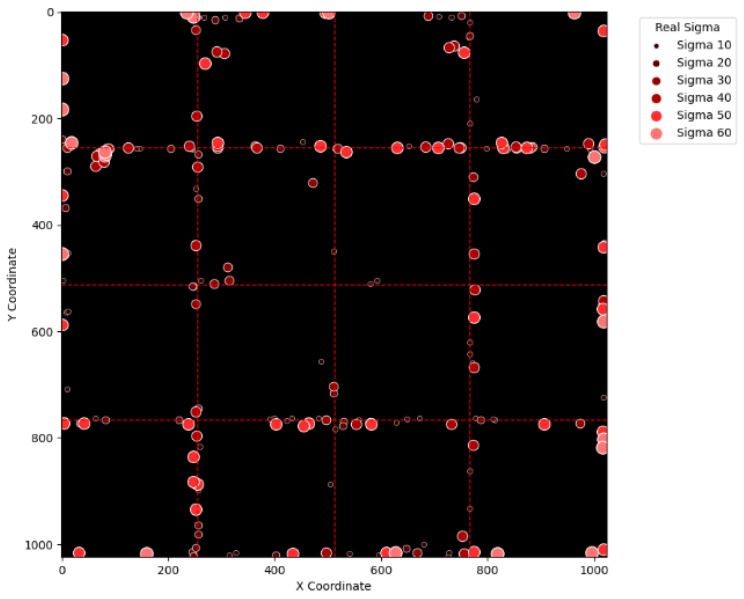
The locations of erroneous cases corresponding to the sigma values varying from 10 to 60 in Table 5. (Maximum intensity = $10^{6}$ ADC and error ≡ +/−4 pixels.)

**Figure 22 sensors-25-04985-f022:**
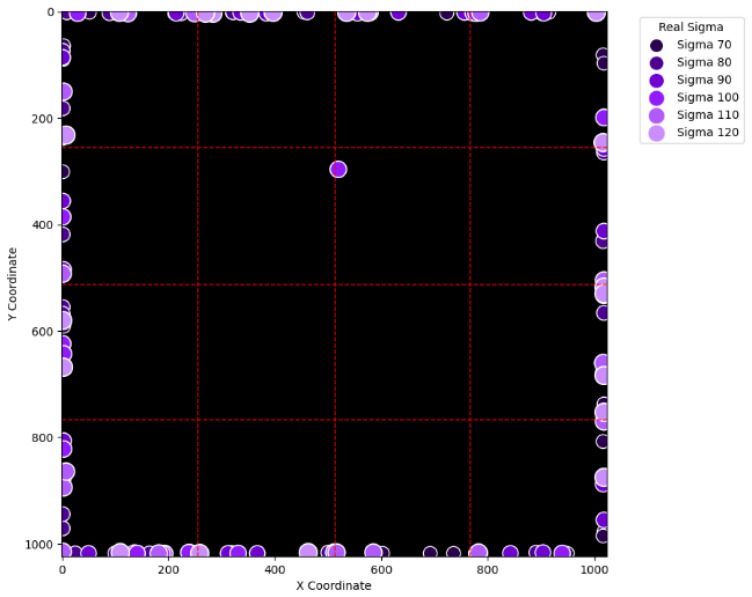
The locations of erroneous cases corresponding to the sigma values varying from 70 to 120 in Table 5. (Maximum intensity = $10^{6}$ ADC and error ≡ +/−4 pixels.)

**Table 1 sensors-25-04985-t001:** Important parameter values of the Boag model adopted by this research study.

*A* (cm^2^)	*E* (V/cm)	*d* (cm)
0.04	800	0.3

**Table 2 sensors-25-04985-t002:** Simulation results with the maximum intensity of 10^4^ ADC with an error definition of more than +/−2 pixels away from the real location of the maximum intensity.

σ	Count	Correct	Error Rate
10	2000	1927	3.65%
20	2000	1961	1.95%
30	2000	1968	1.60%
40	2000	1956	2.20%
50	2000	1931	3.45%
60	2000	1970	1.50%
70	2000	1973	1.35%
80	2000	1967	1.65%
90	2000	1965	1.75%
100	2000	1964	1.80%
110	2000	1961	1.95%
120	2000	1967	1.65%

**Table 3 sensors-25-04985-t003:** Simulation results with the maximum intensity of 10^4^ ADC with an error definition of more than +/−4 pixels away from the real location of the maximum intensity.

σ	Count	Correct	Error Rate
10	2000	1931	3.45%
20	2000	1970	1.50%
30	2000	1979	1.05%
40	2000	1965	1.75%
50	2000	1948	2.60%
60	2000	1985	0.75%
70	2000	1986	0.70%
80	2000	1981	0.95%
90	2000	1983	0.85%
100	2000	1976	1.20%
110	2000	1977	1.15%
120	2000	1980	1.00%

**Table 4 sensors-25-04985-t004:** Simulation results with the maximum intensity of 10^6^ ADC with an error definition of more than +/−2 pixels away from the real location of the maximum intensity.

σ	Count	Correct	Error Rate
10	2000	1921	3.95%
20	2000	1951	2.45%
30	2000	1975	1.25%
40	2000	1943	2.85%
50	2000	1932	3.40%
60	2000	1964	1.80%
70	2000	1959	2.05%
80	2000	1962	1.90%
90	2000	1959	2.05%
100	2000	1966	1.70%
110	2000	1975	1.25%
120	2000	1961	1.95%

**Table 5 sensors-25-04985-t005:** Simulation results with the maximum intensity of 10^6^ ADC with an error definition of more than +/−4 pixels away from the real location of the maximum intensity.

σ	Count	Correct	Error Rate
10	2000	1932	3.40%
20	2000	1965	1.75%
30	2000	1983	0.85%
40	2000	1960	2.00%
50	2000	1957	2.15%
60	2000	1983	0.85%
70	2000	1972	1.40%
80	2000	1974	1.30%
90	2000	1971	1.45%
100	2000	1981	0.95%
110	2000	1984	0.80%
120	2000	1980	1.00%

**Table 6 sensors-25-04985-t006:** The relative errors of the predicted amplitude 10^4^ with different variances of proton Gaussian beams in the virtual layer. Note that one pixel in the virtual layer maps a 0.25 mm dimension, and the simulation times are set to 2000.

Variance (pixels)	σ = 10	σ = 20	σ = 30	σ = 40	σ = 50	σ = 60
Relative error (%)	10.46 ± 3.43	2.19 ± 0.82	0.88 ± 0.37	0.47 ± 0.20	0.29 ± 0.13	0.20 ± 0.09
Variance (pixels)	σ = 70	σ = 80	σ = 90	σ = 100	σ = 110	σ = 120
Relative error (%)	0.15 ± 0.07	0.11 ± 0.05	0.09 ± 0.04	0.07 ± 0.03	0.06 ± 0.03	0.07 ± 0.02

**Table 7 sensors-25-04985-t007:** The relative errors of the predicted amplitude 10^6^ with different variances of proton Gaussian beams in the virtual layer. Note that one pixel in the virtual layer maps a 0.25 mm dimension, and simulation times are set to 2000.

Variance (pixels)	Sigma = 10	Sigma = 20	Sigma = 30	Sigma = 40	Sigma = 50	Sigma = 60
Relative error (%)	10.64 ± 4.31	2.10 ± 0.92	0.87 ± 0.39	0.46 ± 0.22	0.29 ± 0.14	0.20 ± 0.09
Variance (pixels)	Sigma = 70	Sigma = 80	Sigma = 90	Sigma = 100	Sigma = 110	Sigma = 120
Relative error (%)	0.15 ± 0.07	0.11 ± 0.05	0.09 ± 0.04	0.07 ± 0.03	0.06 ± 0.03	0.08 ± 0.02

## Data Availability

The original contributions presented in this study are included in the article. Further inquiries can be directed to the corresponding author.

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
