# Peer review of "Retrieving Proton Beam Information Using Stitching-Based Detector Technique and Intelligent Reconstruction Algorithms"

_sensors, 2025, doi:10.3390/s25164985_

Round 1

Reviewer 1 Report

Comments and Suggestions for Authors
  1. The physical area of Ion Chamber is 2x2 mm2, how could you achieve the 0.25mm resolution discribed in the introduction and abstract? and i found in your reference 28,  the pixel size is 2.5mm, is similar to your design, what's the improvement in your research?
  2. what's the data processing time for the spots measurement?
  3. a very import measurement in the QA is the field analysis , such as a 10*10cm square or even large,  or the spots matrix such as 7*7 in a 25*25cm field, how can you do in this cases?

Reviewer 2 Report

Comments and Suggestions for Authors

Ref.: Manuscript Number: sensors-3748930: Retrieving Proton Beam Information by Using Stitching-Based Detector Technique and Intelligent Reconstruction Algorithms.

Comments to authors

This study presents a proposed intelligent reconstruction algorithm and their excellent performance through simulation. This work also demonstrates the effectiveness of the proposed algorithm using proton beam therapy (PBS) as an example. However, the concept can be easily adapted for use with any other particle-based radiotherapy (RT) techniques, provided the particles used follow a Gaussian distribution.

The manuscript is well structured and well written. However, some improvements are needed to enhance its quality. Therefore, I recommend to see my comments/suggestions below.

1/ In introduction authors write this paragraph  ……..“For example, heavy ions would normally achieve higher relative biological effectiveness (RBE) because of their higher linear energy transfer (LET)”….. Why the LET very high transfer for heavy ions?

2/ The authors need to change the colour of figures 11, 12 and 14, as the colours are very similar.

3/ Authors must explain more, how the 'stitching' process in the SBD technique reconstructs the projected particle beam information, especially in relation to the 128x128 ionisation chambers?

4/ Authors must provide more information on the design of the 25.6 cm² ionisation plate? What are the key considerations for constructing it to achieve the stated resolution of 0.25 mm?

5/ What specific metrics other than the error rate were used to evaluate the effectiveness of the proposed technique beyond the stated 'excellent performance'.

6/ The maximum stated error rate is 3.95%. What factors contribute to this residual and what are the potential clinical implications, even at such a low level, of error in radiotherapy?

7/ What problem does the proposed Stitching-Based Detector (SBD) technique aim to solve in radiotherapy.

8/ Which other types of particle beam in radiotherapy typically follow a Gaussian distribution, making them suitable for this technique?

9/ Authors must explain more, who the beam symmetry contribute to the reconstruction error in the SBD technique.

10/ What probable limitations might arise if the particle beam does not follow a Gaussian distribution.

11/ What about the dose-rate proton beam dosimetry using an ionization chamber.

12/ The results must be compared to others work.

13/ The factor of ion recombination is used to correct the response of an ionization chamber for the lack of complete charge collection. Authors must add the factor expression, and explain more this situation.

Reviewer 3 Report

Comments and Suggestions for Authors

Dear Author,

Please excuse me fot having reviewed your paper with some delay. 

Your paper describes with (too) little detail, in my opinon the approach for calulating the centroid position of the pencil beam. My experience with QA IC for proton-therapy is limited to strip chambers, that faces similar issue in the reconstruction of the pencil beam profile but, do not have the quadrant-ambiguity of you design. 

You approach is interesting because it allows the reduction of ADCs in a very large array. I am aware of other approaches where the ADC has been reduced by channel multiplexing and serialization of conversion. This approach has the drawback of incresing conversion time.

Coming to the paper:

1) I do not understand how the centroid detection works (quadrant in which the beam is). According to what you write, the four ABDC planes are summe and there is no possibility in understanding which of the four quadrants has been hit. I kindly, ask, to better explain that, eventually reducing the large space dedicated to the introduction. 

Best regards

Round 2

Reviewer 1 Report

Comments and Suggestions for Authors

i think the answer is OK for me. 

Reviewer 3 Report

Comments and Suggestions for Authors

Dear Authors,

Thank you for adding information to the paper. I understand, from your text between rows 439 and 457 that the ambiguity resolution in centroid position calculation is based on the position of "halo" of the beam that changes depending on which sub-matrix is hit by the beam. 

You write:  (row 439)At first glance, it may be tempting to think that the data sensed by the ADCs for the four scenarios should be identical. This misconception results mainly from two observations. First, areas “A,” “B,” “C,” and “D” in the four sub-matrices share the same sets of ADCs. Second, the four proton beams are projected onto the intersection of A, B, C, and D in the four quadrants. Despite the misleading first-glance impression, the simulation results presented through the images in the right column of Figure 10 show that the data sensed by the 64 x 64 ADCs demonstrates different behaviors.

My comment: unfortunately you do not explain why this phenomenon occurs. Can you explain why this happens ? What happens if the beam does not hit the borders ? Are you still able to resolve the ambiguity ?

In order to make clearer the paper, I would also suggest to identify ABDC regions in figures: 6, 8, 10, 13
